# Meta-Learning Probabilistic Inference For Prediction

**Jonathan Gordon,**[*] **John Bronskill**[*]
University of Cambridge
{jg801,jfb54}@cam.ac.uk

**Matthias Bauer**
University of Cambridge
Max Planck Institute for Intelligent Systems
bauer@tue.mpg.de

**Sebastian Nowozin**[†]
Google AI Berlin
nowozin@google.com

**Richard E. Turner**
University of Cambridge
Microsoft Research
ret26@cam.ac.uk

## Abstract

This paper introduces a new framework for data efficient and versatile learning. Specifically: 1) We develop ML-PIP, a general framework for Meta-Learning approximate Probabilistic Inference for Prediction. ML-PIP extends existing probabilistic interpretations of meta-learning to cover a broad class of methods. 2) We introduce VERSA, an instance of the framework employing a flexible and versatile amortization network that takes few-shot learning datasets as inputs, with arbitrary numbers of shots, and outputs a distribution over task-specific parameters in a single forward pass. VERSA substitutes optimization at test time with forward passes through inference networks, amortizing the cost of inference and relieving the need for second derivatives during training. 3) We evaluate VERSA on benchmark datasets where the method sets new state-of-the-art results, handles arbitrary numbers of shots, and for classification, arbitrary numbers of classes at train and test time. The power of the approach is then demonstrated through a challenging few-shot ShapeNet view reconstruction task.

## 1 Introduction

Many applications require predictions to be made on myriad small, but related datasets. In such cases, it is natural to desire learners that can rapidly adapt to new datasets at test time. These applications have given rise to vast interest in *few-shot learning* (Fei-Fei et al., 2006; Lake et al., 2011), which emphasizes *data efficiency* via information sharing across related tasks. Despite recent advances, notably in meta-learning based approaches (Ravi and Larochelle, 2017; Vinyals et al., 2016; Edwards and Storkey, 2017; Finn et al., 2017; Lacoste et al., 2018), there remains a lack of general purpose methods for flexible, data-efficient learning.

Due to the ubiquity of recent work, a unifying view is needed to understand and improve these methods. Existing frameworks (Grant et al., 2018; Finn et al., 2018) are limited to specific families of approaches. In this paper we develop a framework for meta-learning approximate probabilistic inference for prediction (ML-PIP), providing this view in terms of amortizing posterior predictive distributions. In Section 4, we show that ML-PIP re-frames and extends existing point-estimate probabilistic interpretations of meta-learning (Grant et al., 2018; Finn et al., 2018) to cover a broader class of methods, including gradient based meta-learning (Finn et al., 2017; Ravi and Larochelle, 2017), metric based meta-learning (Snell et al., 2017), amortized MAP inference (Qiao et al., 2017) and conditional probability modelling (Garnelo et al., 2018a;b).

The framework incorporates three key elements. First, we leverage shared statistical structure between tasks via hierarchical probabilistic models developed for multi-task and transfer learning (Heskes,

---

[*] Authors contributed equally

[†] Work done while at Microsoft Research

2000; Bakker and Heskes, 2003). Second, we share information between tasks about how to learn and perform inference using meta-learning (Naik and Mammone, 1992; Thrun and Pratt, 2012; Schmidhuber, 1987). Since uncertainty is rife in small datasets, we provide a procedure for meta-learning probabilistic inference. Third, we enable fast learning that can flexibly handle a wide range of tasks and learning settings via amortization (Kingma and Welling, 2014; Rezende et al., 2014).

Building on the framework, we propose a new method – VERSA – which substitutes optimization procedures at test time with forward passes through inference networks. This amortizes the cost of inference, resulting in faster test-time performance, and relieves the need for second derivatives during training. VERSA employs a flexible amortization network that takes few-shot learning datasets, and outputs a distribution over task-specific parameters in a single forward pass. The network can handle arbitrary numbers of shots, and for classification, arbitrary numbers of classes at train and test time (see Section 3). In Section 5, we evaluate VERSA on (i) standard benchmarks where the method sets new state-of-the-art results, (ii) settings where test conditions (shot and way) differ from training, and (iii) a challenging one-shot view reconstruction task.

## 2 META-LEARNING PROBABILISTIC INFERENCE FOR PREDICTION

We now present the framework that consists of (i) a multi-task probabilistic model, and (ii) a method for meta-learning probabilistic inference.

### 2.1 PROBABILISTIC MODEL

Two principles guide the choice of model. First, the use of discriminative models to maximize predictive performance on supervised learning tasks (Ng and Jordan, 2002). Second, the need to leverage shared statistical structure between tasks (i.e. multi-task learning). These criteria are met by the standard multi-task directed graphical model shown in Fig. 1 that employs shared parameters $\theta$, which are common to all tasks, and task specific parameters $\{\psi^{(t)}\}_{t=1}^T$. Inputs are denoted $x$ and outputs $y$. Training data $D^{(t)} = \{(x_n^{(t)}, y_n^{(t)})\}_{n=1}^{N_t}$, and test data $\{(\tilde{x}_m^{(t)}, \tilde{y}_m^{(t)})\}_{m=1}^{M_t}$ are explicitly distinguished for each task $t$, as this is key for few-shot learning.

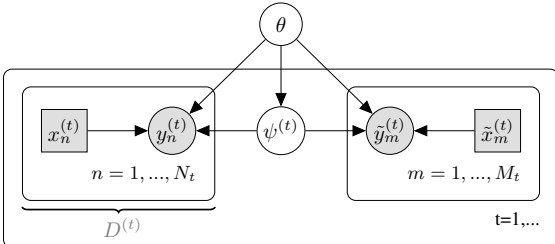

**Figure 1:** Directed graphical model for multi-task learning.

Let $X^{(t)}$ and $Y^{(t)}$ denote all the inputs and outputs (both test and train) for task $t$. The joint probability of the outputs and task specific parameters for $T$ tasks, given the inputs and global parameters is:

$$p\left(\{Y^{(t)}, \psi^{(t)}\}_{t=1}^T | \{X^{(t)}\}_{t=1}^T, \theta\right) = \prod_{t=1}^T p\left(\psi^{(t)}|\theta\right) \prod_{n=1}^{N_t} p\left(y_n^{(t)}|x_n^{(t)}, \psi^{(t)}, \theta\right) \prod_{m=1}^{M_t} p\left(\tilde{y}_m^{(t)}|\tilde{x}_m^{(t)}, \psi^{(t)}, \theta\right).$$

In the next section, the goal is to meta-learn fast and accurate approximations to the posterior predictive distribution $p(\tilde{y}^{(t)}|\tilde{x}^{(t)}, \theta) = \int p(\tilde{y}^{(t)}|\tilde{x}^{(t)}, \psi^{(t)}, \theta)p(\psi^{(t)}|\tilde{x}^{(t)}, D^{(t)}, \theta)\mathrm{d}\psi^{(t)}$ for unseen tasks $t$.

### 2.2 PROBABILISTIC INFERENCE

This section provides a framework for meta-learning approximate inference that is a simple reframing and extension of existing approaches (Finn et al., 2017; Grant et al., 2018). We will employ point estimates for the shared parameters $\theta$ since data across all tasks will pin down their value. Distributional estimates will be used for the task-specific parameters since only a few shots constrain them.

Once the shared parameters are learned, the probabilistic solution to few-shot learning in the model above comprises two steps. First, form the posterior distribution over the task-specific parameters $p(\psi^{(t)}|\tilde{x}^{(t)}, D^{(t)}, \theta)$. Second, compute the posterior predictive $p(\tilde{y}^{(t)}|\tilde{x}^{(t)}, \theta)$. These steps will require approximation and the emphasis here is on performing this quickly at test time. We will describe the form of the approximation, the optimization problem used to learn it, and how to implement this efficiently below. In what follows we initially suppress dependencies on the inputs $\tilde{x}$ and shared parameters $\theta$ to reduce notational clutter, but will reintroduce these at the end of the section.

**Specification of the approximate posterior predictive distribution.**  Our framework approximates the posterior predictive distribution by an amortized distribution $q_\phi(\tilde{y}|D)$. That is, we learn a feed-forward inference network with parameters $\phi$ that takes any training dataset $D^{(t)}$ and test input $\tilde{x}$ as inputs and returns the predictive distribution over the test output $\tilde{y}^{(t)}$. We construct this by amortizing the approximate posterior $q_\phi(\psi|D)$ and then form the approximate posterior predictive distribution using:

$$q_\phi(\tilde{y}|D) = \int p(\tilde{y}|\psi)q_\phi(\psi|D)\mathrm{d}\psi. \tag{1}$$

This step may require additional approximation e.g. Monte Carlo sampling. The amortization will enable fast predictions at test time. The form of these distributions is identical to those used in amortized variational inference (Edwards and Storkey, 2017; Kingma and Welling, 2014). In this work, we use a factorized Gaussian distribution for $q_\phi(\psi|D^{(t)})$ with means and variances set by the amortization network. However, the training method described next is different.

**Meta-learning the approximate posterior predictive distribution.**  The quality of the approximate posterior predictive for a single task will be measured by the KL-divergence between the true and approximate posterior predictive distribution $\mathrm{KL}\left[p(\tilde{y}|D)\|q_\phi(\tilde{y}|D)\right]$. The goal of learning will be to minimize the expected value of this KL averaged over tasks,

$$\phi^* = \arg\min_\phi \mathop{\mathbb{E}}_{p(D)}\left[\mathrm{KL}\left[p(\tilde{y}|D)\|q_\phi(\tilde{y}|D)\right]\right] = \arg\max_\phi \mathop{\mathbb{E}}_{p(\tilde{y},D)}\left[\log\int p(\tilde{y}|\psi)q_\phi(\psi|D)\mathrm{d}\psi\right]. \tag{2}$$

Training will therefore return parameters $\phi$ that best approximate the posterior predictive distribution in an average KL sense. So, if the approximate posterior $q_\phi(\psi|D)$ is rich enough, *global* optimization will recover the true posterior $p(\psi|D)$ (assuming $p(\psi|D)$ obeys identifiability conditions (Casella and Berger, 2002)).[1] Thus, the amortized procedure meta-learns approximate inference that supports accurate prediction. Appendix A provides a generalized derivation of the framework, grounded in Bayesian decision theory (Jaynes, 2003).

The right hand side of Eq. (2) indicates how training could proceed: (i) select a task $t$ at random, (ii) sample some training data $D^{(t)}$, (iii) form the posterior predictive $q_\phi(\cdot|D^{(t)})$ and, (iv) compute the log-density $\log q_\phi(\tilde{y}^{(t)}|D^{(t)})$ at test data $\tilde{y}^{(t)}$ *not included in* $D^{(t)}$. Repeating this process many times and averaging the results would provide an unbiased estimate of the objective which can then be optimized. This perspective also makes it clear that the procedure is scoring the approximate inference procedure by simulating approximate Bayesian held-out log-likelihood evaluation. Importantly, while an inference network is used to approximate posterior distributions, the training procedure differs significantly from standard variational inference. In particular, rather than minimizing $\mathrm{KL}(q_\phi(\psi|D)\|p(\psi|D))$, our objective function directly focuses on the posterior predictive distribution and minimizes $\mathrm{KL}(p(\tilde{y}|D)\|q_\phi(\tilde{y}|D))$.

**End-to-end stochastic training.**  Armed by the insights above we now layout the full training procedure. We reintroduce inputs and shared parameters $\theta$ and the objective becomes:

$$\mathcal{L}(\phi) = -\mathop{\mathbb{E}}_{p(D,\tilde{y},\tilde{x})}\left[\log q_\phi(\tilde{y}|\tilde{x},\theta)\right] = -\mathop{\mathbb{E}}_{p(D,\tilde{y},\tilde{x})}\left[\log\int p(\tilde{y}|\tilde{x},\psi,\theta)q_\phi(\psi|D,\theta)\mathrm{d}\psi\right]. \tag{3}$$

We optimize the objective over the shared parameters $\theta$ as this will maximize predictive performance (i.e., Bayesian held out likelihood). An end-to-end stochastic training objective for $\theta$ and $\phi$ is:

$$\hat{\mathcal{L}}(\theta,\phi) = \frac{1}{MT}\sum_{M,T}\log\frac{1}{L}\sum_{l=1}^{L}p\left(\tilde{y}_m^{(t)}|\tilde{x}_m^{(t)},\psi_l^{(t)},\theta\right), \quad \text{with } \psi_l^{(t)} \sim q_\phi(\psi|D^{(t)},\theta) \tag{4}$$

---

[1]Note that the true *predictive* posterior $p(y|D)$ is recovered regardless of the identifiability of $p(\psi|D)$.

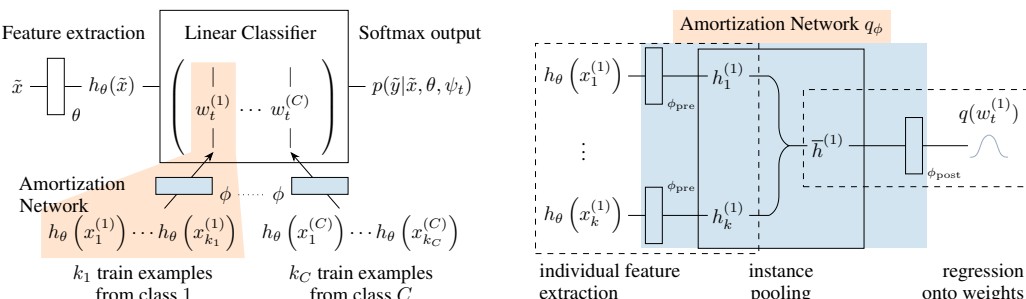

**Figure 2:** Computational flow of VERSA for few-shot classification with the context-independent approximation. *Left:* A test point $\tilde{x}$ is mapped to its softmax output through a feature extractor neural network and a linear classifier (fully connected layer). The global parameters $\theta$ of the feature extractor are shared between tasks whereas the weight vectors $w_t^{(c)}$ of the linear classifier are task specific and inferred through an amortization network with parameters $\phi$. *Right:* Amortization network that maps the extracted features of the $k$ training examples of a particular class to the corresponding weight vector of the linear classifier.

and $\{\tilde{y}_m^{(t)}, \tilde{x}_m^{(t)}, D^{(t)}\} \sim p(\tilde{y}, \tilde{x}, D)$, where $p$ represents the data distribution (e.g., sampling tasks and splitting them into disjoint training data $D$ and test data $\{(\tilde{x}_m^{(t)}, \tilde{y}_m^{(t)})\}_{m=1}^{M_t}$). This type of training therefore uses episodic train / test splits at meta-train time. We have also approximated the integral over $\psi$ using $L$ Monte Carlo samples. The local reparametrization (Kingma et al., 2015) trick enables optimization. Interestingly, the learning objective does not require an explicit specification of the prior distribution over parameters, $p(\psi^{(t)}|\theta)$, learning it implicitly through $q_\phi(\psi|D, \theta)$ instead.

In summary, we have developed an approach for Meta-Learning Probabilistic Inference for Prediction (ML-PIP). A simple investigation of the inference method with synthetic data is provided in Section 5.1. In Section 4 we will show that this formulation unifies a number of existing approaches, but first we discuss a particular instance of the ML-PIP framework that supports versatile learning.

## 3 VERSATILE AMORTIZED INFERENCE

A versatile system is one that makes inferences both rapidly *and* flexibly. By rapidly we mean that test-time inference involves only simple computation such as a feed-forward pass through a neural network. By flexibly we mean that the system supports a variety of tasks – including variable numbers of shots or numbers of classes in classification problems – without retraining. Rapid inference comes automatically with the use of a deep neural network to amortize the approximate posterior distribution $q$. However, it typically comes at the cost of flexibility: amortized inference is usually limited to a single specific task. Below, we discuss design choices that enable us to retain flexibility.

**Inference with sets as inputs.** The amortization network takes data sets of variable size as inputs whose ordering we should be invariant to. We use permutation-invariant *instance-pooling* operations to process these sets similarly to Qi et al. (2017) and as formalized in Zaheer et al. (2017). The instance-pooling operation ensures that the network can process any number of training observations.

**VERSA for Few-Shot Image Classification.** For few-shot image classification, our parameterization of the probabilistic model is inspired by early work from Heskes (2000); Bakker and Heskes (2003) and recent extensions to deep learning (Bauer et al., 2017; Qiao et al., 2017). A feature extractor neural network $h_\theta(x) \in \mathbb{R}^{d_\theta}$, shared across all tasks, feeds into a set of task-specific linear classifiers with softmax outputs and weights and biases $\psi^{(t)} = \{W^{(t)}, b^{(t)}\}$ (see Fig. 2).

A naive amortization requires the approximate posterior $q_\phi(\psi|D, \theta)$ to model the distribution over full weight matrices in $\mathbb{R}^{d_\theta \times C}$ (and biases). This requires the specification of the number of few-shot classes $C$ ahead of time and limits inference to this chosen number. Moreover, it is difficult to meta-learn systems that directly output large matrices as the output dimensionality is high. We therefore propose specifying $q_\phi(\psi|D, \theta)$ in a *context independent* manner such that each weight vector $\psi_c$ depends only on examples from class $c$, by amortizing individual weight vectors associated with a

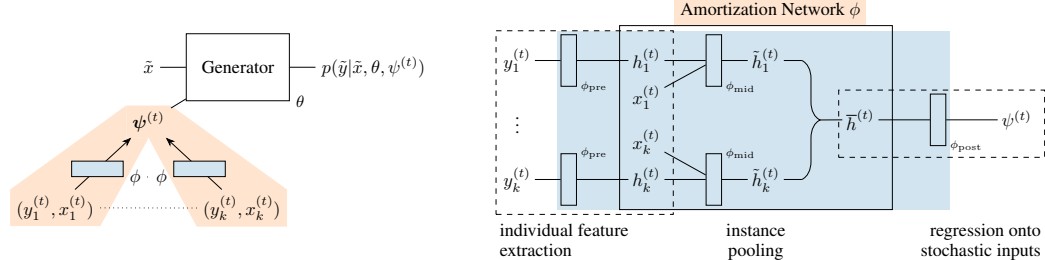

**Figure 3:** Computational flow of VERSA for few-shot view reconstruction. *Left:* A set of training images and angles $\{(y_n^{(t)}, x_n^{(t)})\}_{n=1}^k$ are mapped to a stochastic input $\psi^{(t)}$ through the amortization network $q_\phi$. $\psi^{(t)}$ is then concatenated with a test angle $\tilde{x}$ and mapped onto a new image through the generator $\theta$. *Right:* Amortization network that maps $k$ image/angle examples of a particular object-instance to the corresponding stochastic input.

single softmax output instead of the entire weight matrix directly. To reduce the number of learned parameters, the amortization network operates directly on the extracted features $h_\theta(x)$:

$$q_\phi(\psi|D, \theta) = \prod_{c=1}^{C} q_\phi\left(\psi_c | \{h_\theta(x_n^c)\}_{n=1}^{k_c}, \theta\right). \tag{5}$$

Note that in our implementation, end-to-end training is employed, i.e., we backpropagate to $\theta$ through the inference network. Here $k_c$ is the number of observed examples in class $c$ and $\psi_c = \{w_c, b_c\}$ denotes the weight vector and bias of the linear classifier associated with that class. Thus, we construct the classification matrix $\psi^{(t)}$ by performing $C$ feed-forward passes through the inference network $q_\phi(\psi|D, \theta)$ (see Fig. 2).

The assumption of context independent inference is an approximation. In Appendix B, we provide theoretical and empirical justification for its validity. Our theoretical arguments use insights from Density Ratio Estimation (Mohamed, 2018; Sugiyama et al., 2012), and we empirically demonstrate that full approximate posterior distributions are close to their context independent counterparts. Critically, the context independent approximation addresses all the limitations of a naive amortization mentioned above: (i) the inference network needs to amortize far fewer parameters whose number does not scale with number of classes $C$ (a single weight vector instead of the entire matrix); (ii) the amortization network can be meta-trained with different numbers of classes per task, and (iii) the number of classes $C$ can vary at test-time.

**VERSA for Few-Shot Image Reconstruction (Regression).**  We consider a challenging few-shot learning task with a complex (high dimensional and continuous) output space. We define view reconstruction as the ability to infer how an object looks from any desired angle based on a small set of observed views. We frame this as a multi-output regression task from a set of training images with known orientations to output images with specified orientations.

Our generative model is similar to the generator of a GAN or the decoder of a VAE: A latent vector $\psi^{(t)} \in \mathbb{R}^{d_\psi}$, which acts as an object-instance level input to the generator, is concatenated with an angle representation and mapped through the generator to produce an image at the specified orientation. In this setting, we treat all parameters $\theta$ of the generator network as global parameters (see Appendix E.1 for full details of the architecture), whereas the latent inputs $\psi^{(t)}$ are the task-specific parameters. We use a Gaussian likelihood in pixel space for the outputs of the generator. To ensure that the output means are between zero and one, we use a sigmoid activation after the final layer. $\phi$ parameterizes an amortization network that first processes the image representations of an object, concatenates them with their associated view orientations, and processes them further before instance-pooling. From the pooled representations, $q_\phi(\psi|D, \theta)$ produces a distribution over vectors $\psi^{(t)}$. This process is illustrated in Fig. 3.

## 4    ML-PIP UNIFIES DISPARATE RELATED WORK

In this section, we continue in the spirit of Grant et al. (2018), and recast a broader class of meta-learning approaches as approximate inference in hierarchical models. We show that ML-PIP unifies a number of important approaches to meta-learning, including *both* gradient and metric based variants, as well as amortized MAP inference and conditional modelling approaches (Garnelo et al., 2018a). We lay out these connections, most of which rely on point estimates for the task-specific parameters corresponding to $q(\psi^{(t)}|D^{(t)}, \theta) = \delta\left(\psi^{(t)} - \psi^*(D^{(t)}, \theta)\right)$. In addition, we compare previous approaches to VERSA.

**Gradient-Based Meta-Learning.**    Let the task-specific parameters $\psi^{(t)}$ be all the parameters in a neural network. Consider a point estimate formed by taking a step of gradient ascent of the training loss, initialized at $\psi_0$ and with learning rate $\eta$.

$$\psi^*(D^{(t)}, \theta) = \psi_0 + \eta\frac{\partial}{\partial\psi}\log\sum_{n=1}^{N_t} p(y_n^{(t)}|x_n^{(t)}, \psi, \theta)\bigg|_{\psi_0}. \qquad (6)$$

This is an example of semi-amortized inference (Kim et al., 2018b), as the only shared inference parameters are the initialization and learning rate, and optimization is required for each task (albeit only for one step). Importantly, Eq. (6) recovers *Model-agnostic meta-learning* (Finn et al., 2017), providing a perspective as semi-amortized ML-PIP. This perspective is complementary to that of Grant et al. (2018) who justify the one-step gradient parameter update employed by MAML through MAP inference and the form of the prior $p(\psi|\theta)$. Note that the episodic meta-train / meta-test splits do not fall out of this perspective. Instead we view the update choice as one of amortization which is trained using the predictive KL and naturally recovers the test-train splits. More generally, multiple gradient steps could be fed into an RNN to compute $\psi^*$ which recovers Ravi and Larochelle (2017). In comparison to these methods, besides being distributional over $\psi$, VERSA relieves the need to back-propagate through gradient based updates during training and compute gradients at test time, as well as enables the treatment of both local and global parameters which simplifies inference.

**Metric-Based Few-Shot Learning.**    Let the task-specific parameters be the top layer softmax weights and biases of a neural network $\psi^{(t)} = \{w_c^{(t)}, b_c^{(t)}\}_{c=1}^C$. The shared parameters are the lower layer weights. Consider amortized point estimates for these parameters constructed by averaging the top-layer activations for each class,

$$\psi^*(D^{(t)}, \theta) = \{w_c^*, b_c^*\}_{c=1}^C = \left\{\mu_c^{(t)}, -\|\mu_c^{(t)}\|^2/2\right\}_{c=1}^C \quad \text{where} \quad \mu_c^{(t)} = \frac{1}{k_c}\sum_{n=1}^{k_c} h_\theta(x_n^{(c)}) \qquad (7)$$

These choices lead to the following predictive distribution:

$$p(\tilde{y}^{(t)} = c|\tilde{x}^{(t)}, \theta) \propto \exp\left(-d(h_\theta(\tilde{x}^{(t)}), \mu_c^{(t)})\right) = \exp\left(h_\theta(\tilde{x}^{(t)})^T\mu_c^{(t)} - \frac{1}{2}\|\mu_c^{(t)}\|^2\right), \qquad (8)$$

which recovers *prototypical networks* (Snell et al., 2017) using a Euclidean distance function $d$ with the final hidden layer being the embedding space. In comparison, VERSA is distributional and it uses a more flexible amortization function that goes beyond averaging of activations.

**Amortized MAP inference.**    Qiao et al. (2017) proposed a method for predicting weights of classes from activations of a pre-trained network to support i) online learning on a single task to which new few-shot classes are incrementally added, ii) transfer from a high-shot classification task to a separate low-shot classification task. This is an example usage of hyper-networks (Ha et al., 2016) to amortize learning about weights, and can be recovered by the ML-PIP framework by pre-training $\theta$ and performing MAP inference for $\psi$. VERSA goes beyond point estimates and although its amortization network is similar in spirit, it is more general, employing end-to-end training and supporting full multi-task learning by sharing information between many tasks.

**Conditional models trained via maximum likelihood.**    In cases where a point estimate of the task-specific parameters are used the predictive becomes

$$q_\phi(\tilde{y}|D, \theta) = \int p(\tilde{y}|\psi, \theta)q_\phi(\psi|D, \theta)\mathrm{d}\psi = p(\tilde{y}|\psi^*(D, \theta), \theta). \qquad (9)$$

In such cases the amortization network that computes $\psi^*(D, \theta)$ can be equivalently viewed as part of the model specification rather than the inference scheme. From this perspective, the ML-PIP training procedure for $\phi$ and $\theta$ is equivalent to training a conditional model $p(\tilde{y}|\psi_\phi^*(D, \theta), \theta)$ via maximum likelihood estimation, establishing a strong connection to neural processes (Garnelo et al., 2018a;b).

**Comparison to Variational Inference (VI).** Standard application of amortized VI (Kingma and Welling, 2014; Rezende et al., 2014; Kingma et al., 2015; Blundell et al., 2015) for $\psi$ in the multi-task discriminative model optimizes the Monte Carlo approximated free-energy w.r.t. $\phi$ and $\theta$:

$$\hat{\mathcal{L}}(\theta, \phi) = \frac{1}{T} \sum_{t=1}^{T} \left( \sum_{(x,y) \in D^{(t)}} \left( \frac{1}{L} \sum_{l=1}^{L} \log p(y^{(t)}|x^{(t)}, \psi_l^{(t)}, \theta) \right) - \text{KL} \left[ q_\phi(\psi|D^{(t)}, \theta) \| p(\psi|\theta) \right] \right),$$
(10)

where $\psi_l^{(t)} \sim q_\phi(\psi|D^{(t)}, \theta)$. In addition to the conceptual difference from ML-PIP (discussed in Section 2.1), this differs from the ML-PIP objective by i) not employing meta train / test splits, and ii) including the KL for regularization instead. In Section 5, we show that VERSA significantly improves over standard VI in the few-shot classification case and compare to recent VI/meta-learning hybrids.

## 5 EXPERIMENTS AND RESULTS

We evaluate VERSA on several few-shot learning tasks. We begin with toy experiments to investigate the properties of the amortized posterior inference achieved by VERSA. We then report few-shot classification results using the Omniglot and *mini*ImageNet datasets in Section 5.2, and demonstrate VERSA's ability to retain high accuracy as the shot and way are varied at test time. In Section 5.3, we examine VERSA's performance on a one-shot view reconstruction task with ShapeNet objects.[2]

### 5.1 POSTERIOR INFERENCE WITH TOY DATA

To investigate the approximate inference performed by our training procedure, we run the following experiment. We first generate data from a Gaussian distribution with a mean that varies across tasks:

$$p(\theta) = \delta(\theta - 0); \quad p\left(\psi^{(t)}|\theta\right) = \mathcal{N}\left(\psi^{(t)}; \theta, \sigma_\psi^2\right); \quad p\left(y_n^{(t)}|\psi^{(t)}\right) = \mathcal{N}\left(y_n^{(t)}; \psi^{(t)}, \sigma_y^2\right). \quad (11)$$

We generate $T = 250$ tasks in two separate experiments, having $N \in \{5, 10\}$ train observations and $M = 15$ test observations. We introduce the inference network $q_\phi(\psi|D^{(t)}) = \mathcal{N}(\psi; \mu_q^{(t)}, \sigma_q^{(t)2})$, amortizing inference as:

$$\mu_q^{(t)} = w_\mu \sum_{n=1}^{N} y_n^{(t)} + b_\mu, \quad \sigma_q^{(t)2} = \exp\left(w_\sigma \sum_{n=1}^{N} y_n^{(t)} + b_\sigma\right). \quad (12)$$

The learnable parameters $\phi = \{w_\mu, b_\mu, w_\sigma, b_\sigma\}$ are trained with the objective function in Eq. (4). The model is trained to convergence with Adam (Kingma and Ba, 2015) using mini-batches of tasks from the generated dataset. Then, a separate set of tasks is generated from the same generative process, and the posterior $q_\phi(\psi|D)$ is inferred with the learned amortization parameters. The true posterior over $\psi$ is Gaussian with a mean that depends on the task, and may be computed analytically. Fig. 4 shows the approximate posterior distributions inferred for unseen test sets by the trained amortization networks. The evaluation shows that the inference procedure is able to recover accurate posterior distributions over $\psi$, despite minimizing a predictive KL divergence in data space.

### 5.2 FEW-SHOT CLASSIFICATION

We evaluate VERSA on standard few-shot classification tasks in comparison to previous work. Specifically, we consider the Omniglot (Lake et al., 2011) and *mini*ImageNet (Ravi and Larochelle, 2017) datasets which are $C$-way classification tasks with $k_c$ examples per class. VERSA follows the implementation in Sections 2 and 3, and the approximate inference scheme in Eq. (5). We follow the experimental protocol established by Vinyals et al. (2016) for Omniglot and Ravi and Larochelle

---

[2]Source code for the experiments is available at `https://github.com/Gordonjo/versa`.

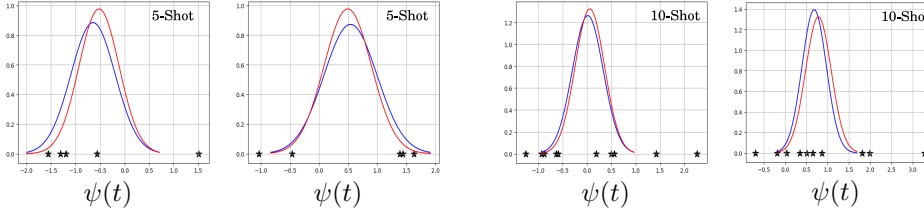

**Figure 4:** True posteriors $p(\psi|D)$ (——) and approximate posteriors $q_\phi(\psi|D)$ (——) for unseen test sets (⋆) in the experiment. In both cases (five and ten shot), the approximate posterior closely resembles the true posterior given the observed data.

(2017) for *mini*Imagenet, using equivalent architectures for $h_\theta$. Training is carried out in an episodic manner: for each task, $k_c$ examples are used as training inputs to infer $q_\phi(\psi^{(c)}|D, \theta)$ for each class, and an additional set of examples is used to evaluate the objective function. Full details of data preparation and network architectures are provided in Appendix D.

Table 3 details few-shot classification performance for VERSA as well as competitive approaches. The tables include results for only those approaches with comparable training procedures and convolutional feature extraction architectures. Approaches that employ pre-training and/or residual networks (Bauer et al., 2017; Qiao et al., 2017; Rusu et al., 2018; Gidaris and Komodakis, 2018; Oreshkin et al., 2018; Garcia and Bruna, 2017; Lacoste et al., 2018) have been excluded so that the quality of the learning algorithm can be assessed separately from the power of the underlying discriminative model.

For Omniglot, the training, validation, and test splits have not been specified for previous methods, affecting the comparison. VERSA achieves a new state-of-the-art results (67.37% - up 1.38% over the previous best) on 5-way - 5-shot classification on the *mini*ImageNet benchmark and (97.66% - up 0.02%) on the 20-way - 1 shot Omniglot benchmark for systems using a convolution-based network architecture and an end-to-end training procedure. VERSA is within error bars of state-of-the-art on three other benchmarks including 5-way - 1-shot *mini*ImageNet, 5-way - 5-shot Omniglot, and 5-way - 1-shot Omniglot. Results on the Omniglot 20 way - 5-shot benchmark are very competitive with, but lower than other approaches. While most of the methods evaluated in Table 3 adapt all of the learned parameters for new tasks, VERSA is able to achieve state-of-the-art performance despite adapting only the weights of the top-level classifier.

**Comparison to standard and amortized VI.** To investigate the performance of our inference procedure, we compare it in terms of log-likelihood (Table 1) and accuracy (Table 3) to training the same model using both amortized and non-amortized VI (i.e., Eq. (10)). Derivations and further experimental details are provided in Appendix C. VERSA improves substantially over amortized VI even though the same amortization network is used for both. This is due to VI's tendency to under-fit, especially for small numbers of data points (Trippe and Turner, 2018; Turner and Sahani, 2011) which is compounded when using inference networks (Cremer et al., 2018). Using non-amortized VI

**Table 1:** Negative Log-likelihood (NLL) results for different few-shot settings on Omniglot and *mini*ImageNet. The ± sign indicates the 95% confidence interval over tasks using a Student's t-distribution approximation.

| | *Omniglot* | | | | *miniImageNet* | |
| | 5-way NLL | | 20-way NLL | | 5-way NLL | |
| Method | 1-shot | 5-shot | 1-shot | 5-shot | 1-shot | 5-shot |
|---|---|---|---|---|---|---|
| Amortized VI | $0.179 \pm 0.009$ | $0.137 \pm 0.004$ | $0.456 \pm 0.010$ | $0.253 \pm 0.004$ | $1.328 \pm 0.024$ | $1.165 \pm 0.010$ |
| Non-Amortized VI | $0.144 \pm 0.005$ | $0.025 \pm 0.001$ | $0.393 \pm 0.005$ | $0.078 \pm 0.002$ | | |
| **VERSA** | $0.010 \pm 0.005$ | $0.007 \pm 0.003$ | $0.079 \pm 0.009$ | $0.031 \pm 0.004$ | $1.183 \pm 0.023$ | $0.859 \pm 0.015$ |

improves performance substantially, but does not reach the level of VERSA and forming the posterior is significantly slower as it requires many forward / backward passes through the network. This is similar in spirit to MAML (Finn et al., 2017), though MAML dramatically reduces the number of required iterations by finding good global initializations e.g., five gradient steps for *mini*ImageNet. This is in contrast to the single forward pass required by VERSA.

**Versatility.** VERSA allows us to vary the number of classes $C$ and shots $k_c$ between training and testing (Eq. (5)). Fig. 5a shows that a model trained for a particular $C$-way retains very high accuracy as $C$ is varied. For example, when VERSA is trained for the 20-Way, 5-Shot condition, at test-time it can handle $C = 100$ way conditions and retain an accuracy of approximately 94%. Fig. 5b shows similar robustness as the number of shots $k_c$ is varied. VERSA therefore demonstrates considerable flexibility and robustness to the test-time conditions, but at the same time it is efficient as it only requires forward passes through the network. The time taken to evaluate 1000 test tasks with a 5-way, 5-shot *mini*ImageNet trained model using MAML (`https://github.com/cbfinn/maml`) is 302.9 seconds whereas VERSA took 53.5 seconds on a NVIDIA Tesla P100-PCIE-16GB GPU. This is more than $5\times$ speed advantage in favor of VERSA while bettering MAML in accuracy by 4.26%.

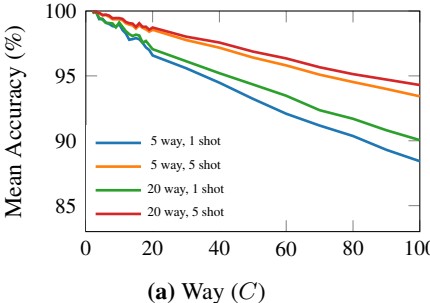

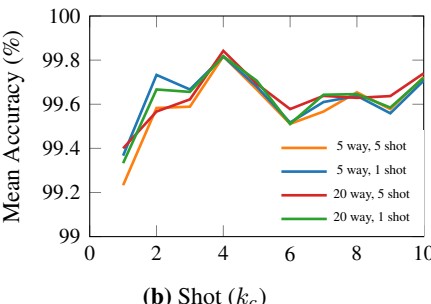

**(a)** Way ($C$)  **(b)** Shot ($k_c$)

**Figure 5:** Test accuracy on Omniglot when varying (a) way (fixing shot to be that used for training) and (b) shot. In Fig. 5b, all models are evaluated on 5-way classification. Colors indicate models trained with different way-shot episodic combinations.

## 5.3 SHAPENET VIEW RECONSTRUCTION

ShapeNetCore v2 (Chang et al., 2015) is a database of 3D objects covering 55 common object categories with ∼51,300 unique objects. For our experiments, we use 12 of the largest object categories. We concatenate all instances from all 12 of the object categories together to obtain a dataset of 37,108 objects. This dataset is then randomly shuffled and we use 70% of the objects for training, 10% for validation, and 20% for testing. For each object, we generate 36 views of size $32 \times 32$ pixels spaced evenly every 10 degrees in azimuth around the object.

We evaluate VERSA by comparing it to a conditional variational autoencoder (C-VAE) with view angles as labels (Kingma et al., 2014; Narayanaswamy et al., 2017) and identical architectures. We train VERSA in an episodic manner and the C-VAE in batch-mode on all 12 object classes at once. We train on a single view selected at random and use the remaining views to evaluate the objective function. For full experimentation details see Appendix E. Fig. 6 shows views of unseen objects from the test set generated from a single shot with VERSA as well as a C-VAE and compares both to ground truth views. Both VERSA and the C-VAE capture the correct orientation of the object in the generated images. However, VERSA produces images that contain much more detail and are visually sharper than the C-VAE images. Although important information is missing due to occlusion in the single shot, VERSA is often able to accurately impute this information presumably due to learning the statistics of these objects. Table 2 provides quantitative comparison results between VERSA with varying shot and the C-VAE. The quantitative metrics all show the superiority of VERSA over a C-VAE. As the number of shots increase to 5, the measurements show a corresponding improvement.

| Model | MSE | SSIM |
|---|---|---|
| C-VAE 1-shot | 0.0269 | 0.5705 |
| VERSA 1-shot | 0.0108 | 0.7893 |
| VERSA 5-shot | 0.0069 | 0.8483 |

**Table 2:** View reconstruction test results. Mean squared error (MSE – lower is better) and the structural similarity index (SSIM - higher is better) (Wang et al., 2004) are measured between the generated and ground truth images. Error bars not shown as they are insignificant.

# 6 CONCLUSIONS

We have introduced ML-PIP, a probabilistic framework for meta-learning. ML-PIP unifies a broad class of recently proposed meta-learning methods, and suggests alternative approaches. Building on ML-PIP, we developed VERSA, a few-shot learning algorithm that avoids the use of gradient based optimization at test time by amortizing posterior inference of task-specific parameters. We evaluated VERSA on several few-shot learning tasks and demonstrated state-of-the-art performance and compelling visual results on a challenging 1-shot view reconstruction task.

**Table 3:** Accuracy results for different few-shot settings on Omniglot and *mini*ImageNet. The $\pm$ sign indicates the 95% confidence interval over tasks using a Student's t-distribution approximation. Bold text indicates the highest scores that overlap in their confidence intervals.

| | Omniglot | | | | miniImageNet | |
| | 5-way accuracy (%) | | 20-way accuracy (%) | | 5-way accuracy (%) | |
| Method | 1-shot | 5-shot | 1-shot | 5-shot | 1-shot | 5-shot |
|---|---|---|---|---|---|---|
| Siamese Nets (Koch et al., 2015) | 97.3 | 98.4 | 88.1 | 97.0 | | |
| Matching Nets (Vinyals et al., 2016) | 98.1 | 98.9 | 93.8 | 98.5 | 46.6 | 60.0 |
| Neural Statistician (Edwards and Storkey, 2017) | 98.1 | 99.5 | 93.2 | 98.1 | | |
| Memory Mod (Kaiser et al., 2017) | 98.4 | 99.6 | 95.0 | 98.6 | | |
| Meta LSTM (Ravi and Larochelle, 2017) | | | | | $43.44 \pm 0.77$ | $60.60 \pm 0.71$ |
| MAML (Finn et al., 2017) | $98.7 \pm 0.4$ | $\mathbf{99.9 \pm 0.1}$ | $95.8 \pm 0.3$ | $98.9 \pm 0.2$ | $48.7 \pm 1.84$ | $63.11 \pm 0.92$ |
| Prototypical Nets[a] (Snell et al., 2017) | 97.4 | 99.3 | 95.4 | 98.7 | $46.61 \pm 0.78$ | $65.77 \pm 0.70$ |
| mAP-SSVM (Triantafillou et al., 2017) | 98.6 | 99.6 | 95.2 | 98.6 | $50.32 \pm 0.80$ | $63.94 \pm 0.72$ |
| mAP-DLM (Triantafillou et al., 2017) | 98.8 | 99.6 | 95.4 | 98.6 | $50.28 \pm 0.80$ | $63.70 \pm 0.70$ |
| LLAMA (Grant et al., 2018) | | | | | $49.40 \pm 1.83$ | |
| PLATIPUS (Finn et al., 2018) | | | | | $50.13 \pm 1.86$ | |
| Meta-SGD (Li et al., 2017) | $\mathbf{99.53 \pm 0.26}$ | $\mathbf{99.93 \pm 0.09}$ | $95.93 \pm 0.38$ | $98.97 \pm 0.19$ | $50.47 \pm 1.87$ | $64.03 \pm 0.94$ |
| SNAIL (Mishra et al., 2018) | $99.07 \pm 0.16$ | $99.78 \pm 0.09$ | $\mathbf{97.64 \pm 0.30}$ | $\mathbf{99.36 \pm 0.18}$ | 45.1 | 55.2 |
| Relation Net (Yang et al., 2018) | $\mathbf{99.6 \pm 0.2}$ | $\mathbf{99.8 \pm 0.1}$ | $\mathbf{97.6 \pm 0.2}$ | $\mathbf{99.1 \pm 0.1}$ | $50.44 \pm 0.82$ | $65.32 \pm 0.70$ |
| Reptile (Nichol and Schulman, 2018) | $97.68 \pm 0.04$ | $99.48 \pm 0.06$ | $89.43 \pm 0.14$ | $97.12 \pm 0.32$ | $49.97 \pm 0.32$ | $\mathbf{65.99 \pm 0.58}$ |
| BMAML (Kim et al., 2018a) | | | | | $\mathbf{53.8 \pm 1.46}$ | |
| Amortized VI | $97.77 \pm 0.55$ | $98.71 \pm 0.22$ | $90.56 \pm 0.54$ | $96.12 \pm 0.23$ | $44.13 \pm 1.78$ | $55.68 \pm 0.91$ |
| Non-Amortized VI | $98.77 \pm 0.18$ | $99.74 \pm 0.06$ | $95.28 \pm 0.19$ | $98.84 \pm 0.09$ | | |
| VERSA (Ours) | $\mathbf{99.70 \pm 0.20}$ | $\mathbf{99.75 \pm 0.13}$ | $\mathbf{97.66 \pm 0.29}$ | $98.77 \pm 0.18$ | $\mathbf{53.40 \pm 1.82}$ | $\mathbf{67.37 \pm 0.86}$ |

[a]We report the performance of Prototypical Networks when training and testing with the same "shot" and "way", which is consistent with the experimental protocol of the other methods listed. We note that Prototypical Networks perform better when trained on higher "way" than that of testing. In particular, when trained on 20-way classification and tested on 5-way, the model achieves $68.20 \pm 0.66\%$.

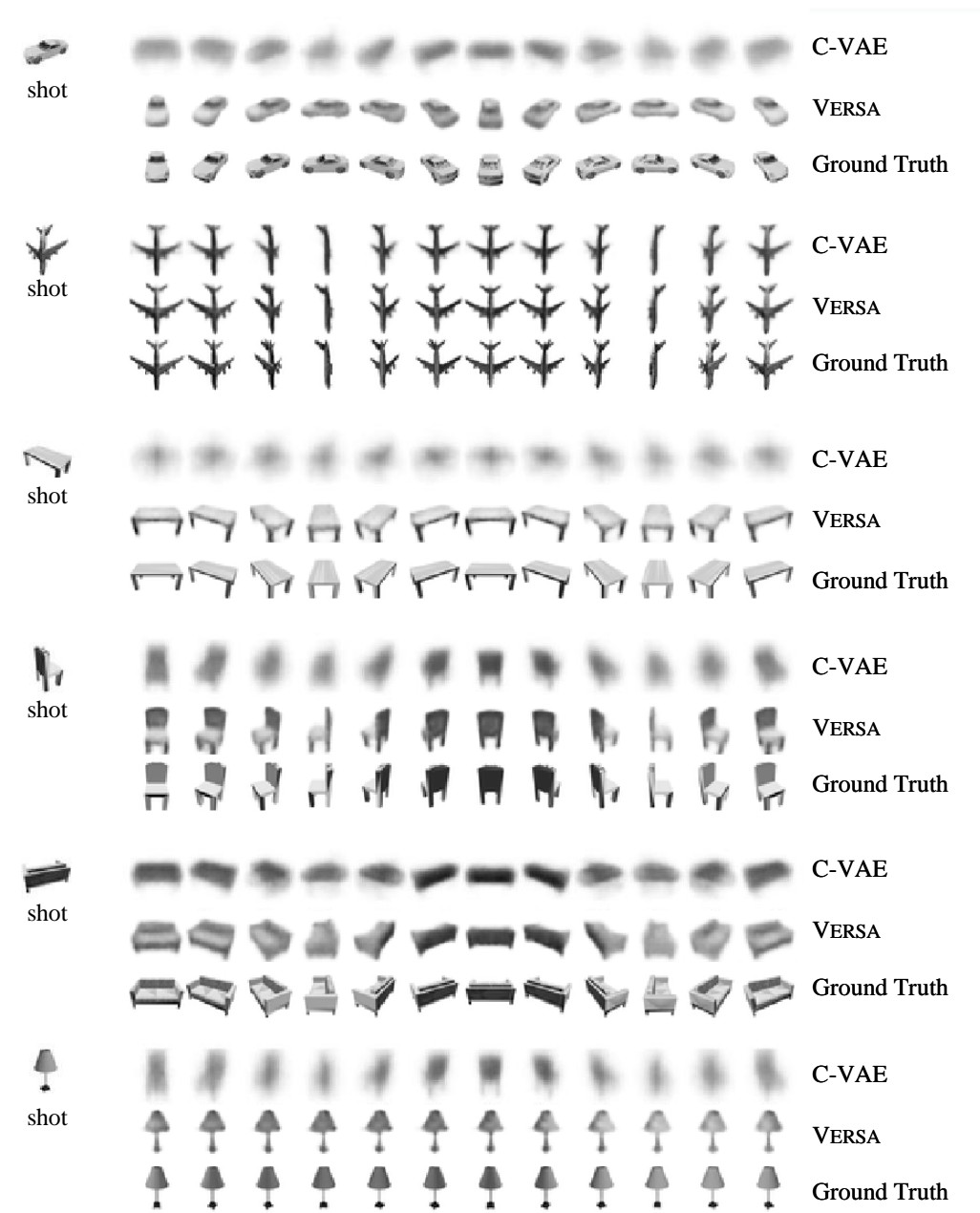

**Figure 6:** Results for ShapeNet view reconstruction for unseen objects from the test set (shown left). The model was trained to reconstruct views from a single orientation. *Top row:* images/views generated by a C-VAE model; *middle row* images/views generated by VERSA; *bottom row:* ground truth images. Views are spaced evenly every 30 degrees in azimuth.

ACKNOWLEDGEMENTS

We thank Ravi and Larochelle for providing the miniImageNet dataset, and Yingzhen Li, Niki Kilbertus, Will Tebbutt, Maria Lomelli, and Robert Pinsler for their useful feedback. J.G. acknowledges funding from a Samsung Doctoral Scholarship. M.B. acknowledges funding by the EPSRC and a Qualcomm European Scholarship in Technology. R.E.T. acknowledges support from EPSRC grants EP/M0269571 and EP/L000776/1.

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

## A BAYESIAN DECISION THEORETIC GENERALIZATION OF ML-PIP

A generalization of the new inference framework presented in Section 2 is based upon Bayesian decision theory (BDT). BDT provides a recipe for making predictions $\hat{y}$ for an unknown test variable $\tilde{y}$ by combining information from observed training data $D^{(t)}$ (here from a single task $t$) and a loss function $L(\tilde{y}, \hat{y})$ that encodes the cost of predicting $\hat{y}$ when the true value is $\tilde{y}$ (Berger, 2013; Jaynes, 2003). In BDT an optimal prediction minimizes the expected loss (suppressing dependencies on the inputs and $\theta$ to reduce notational clutter):[3]

$$\hat{y}^* = \underset{\hat{y}}{\operatorname{argmin}} \int p(\tilde{y}|D^{(t)})L(\tilde{y}, \hat{y})\mathrm{d}\tilde{y}, \quad \text{where} \quad p(\tilde{y}|D^{(t)}) = \int p(\tilde{y}|\psi^{(t)})p(\psi^{(t)}|D^{(t)})\mathrm{d}\psi^{(t)} \tag{A.1}$$

is the Bayesian predictive distribution and $p(\psi^{(t)}|D^{(t)})$ the posterior distribution of $\psi^{(t)}$ given the training data from task $t$.

BDT separates test and training data and so is a natural lens through which to view recent episodic approaches to training that utilize many internal training/test splits (Vinyals et al., 2016). Based on this insight, what follows is a fairly dense derivation of an ultimately simple stochastic variational objective for meta-learning probabilistic inference that is rigorously grounded in Bayesian inference and decision theory.

**Distributional BDT.** We generalize BDT to cases where the goal is to return a full predictive *distribution* $q(\cdot)$ over the unknown test variable $\tilde{y}$ rather than a point prediction. The quality of $q$ is quantified through a distributional loss function $L(\tilde{y}, q(\cdot))$. Typically, if $\tilde{y}$ (the true value of the underlying variable) falls in a low probability region of $q(\cdot)$ the loss will be high, and vice versa. The optimal predictive $q^*$ is found by optimizing the expected distributional loss with $q$ constrained to a distributional family $\mathcal{Q}$:

$$q^* = \underset{q \in \mathcal{Q}}{\operatorname{argmin}} \int p(\tilde{y}|D^{(t)})L(\tilde{y}, q(\cdot))\mathrm{d}\tilde{y}. \tag{A.2}$$

**Amortized variational training.** Here, we amortize $q$ to form quick predictions at test time and learn parameters by minimizing average expected loss *over tasks*. Let $\phi$ be a set of shared variational parameters such that $q(\tilde{y}) = q_\phi(\tilde{y}|D)$ (or $q_\phi$ for shorthand). Now the approximate predictive distribution can take any training dataset $D^{(t)}$ as an argument and directly perform prediction of $\tilde{y}^{(t)}$. The optimal variational parameters are found by minimizing the expected distributional loss across tasks

$$\phi^* = \underset{\phi}{\operatorname{argmin}} \mathcal{L}[q_\phi], \quad \mathcal{L}[q_\phi] = \int p(D)p(\tilde{y}|D)L(\tilde{y}, q_\phi(\cdot|D))\mathrm{d}\tilde{y}\,\mathrm{d}D = \mathbb{E}_{p(D,\tilde{y})}[L(\tilde{y}, q_\phi(\cdot|D))]. \tag{A.3}$$

Here the variables $D, \tilde{x}$ and $\tilde{y}$ are placeholders for integration over all possible datasets, test inputs and outputs. Note that Eq. (A.3) can be stochastically approximated by sampling a task $t$ and randomly partitioning into training data $D$ and test data $\{\tilde{x}_m, \tilde{y}_m\}_{m=1}^M$, which naturally recovers episodic mini-batch training over tasks and data (Vinyals et al., 2016; Ravi and Larochelle, 2017). Critically, this does not require computation of the true predictive distribution. It also emphasizes the meta-learning aspect of the procedure, as the model is *learning how to infer* predictive distributions from training tasks.

**Loss functions.** We employ the log-loss: the negative log density of $q_\phi$ at $\tilde{y}$. In this case,

$$\mathcal{L}[q_\phi] = \mathbb{E}_{p(D,\tilde{y})}[-\log q_\phi(\tilde{y}|D)] = \mathbb{E}_{p(D)}[\mathrm{KL}[p(\tilde{y}|D)\|q_\phi(\tilde{y}|D)] + \mathrm{H}[p(\tilde{y}|D)]], \tag{A.4}$$

where $\mathrm{KL}[p(y)\|q(y)]$ is the KL-divergence, and $\mathrm{H}[p(y)]$ is the entropy of $p$. Eq. (A.4) has the elegant property that the optimal $q_\phi$ is the closest member of $\mathcal{Q}$ (in a KL sense) to the true predictive $p(\tilde{y}|D)$, which is unsurprising as the log-loss is a proper scoring rule (Huszar, 2013). This is reminiscent of the sleep phase in the wake-sleep algorithm (Hinton et al., 1995). Exploration of alternative proper scoring rules (Dawid, 2007) and more task-specific losses (Lacoste-Julien et al., 2011) is left for future work.

---

[3]For discrete outputs the integral may be replaced with a summation.

**Specification of the approximate predictive distribution.** Next, we consider the form of $q_\phi$. Motivated by the optimal predictive distribution, we replace the true posterior by an approximation:

$$q_\phi(\tilde{y}|D) = \int p(\tilde{y}|\psi)q_\phi(\psi|D)\mathrm{d}\psi. \tag{A.5}$$

## B  JUSTIFICATION FOR CONTEXT-INDEPENDENT APPROXIMATION

In this section we lay out both theoretical and empirical justifications for the context-independent approximation detailed in Section 3.

### B.1  THEORETICAL ARGUMENT – DENSITY RATIO ESTIMATION

A principled justification for the approximation is best understood through the lens of density ratio estimation (Mohamed, 2018; Sugiyama et al., 2012). We denote the conditional density of each class as $p(x|y = c)$ and assume equal a priori class probability $p(y = c) = 1/C$. Density ratio theory then uses Bayes' theorem to show that the optimal softmax classifier can be expressed in terms of the conditional densities (Mohamed, 2018; Sugiyama et al., 2012):

$$\text{Softmax}(y = c|x) = \frac{\exp(h(x)^\top w_c)}{\sum_{c'} \exp(h(x)^\top w_{c'})} = p(y = c|x) = \frac{p(x|y = c)}{\sum_{c'} p(x|y = c')}, \tag{B.1}$$

This implies that the optimal classifier will construct estimators for the conditional density for each class, that is $\exp(h(x)^\top w_c) \propto p(x|y = c)$. Importantly for our approximation, notice that these estimates are constructed *independently* for each class, similarly to training a naive Bayes classifier. VERSA mirrors this optimal form using:

$$\log p(x|y = c) \propto h_\theta(\tilde{x})^\top w_c, \tag{B.2}$$

where $w_c \sim q_\phi(w|\{x_n|y_n = c\})$ for each class in a given task. Under ideal conditions (i.e., if one could perfectly estimate $p(\tilde{x}|y = c)$), the context-independent assumption holds, further motivating our design.

### B.2  EMPIRICAL JUSTIFICATION

Here we detail a simple experiment to evaluate the validity of the context-independent inference assumption. The goal of the experiment is to examine if weights may be context-independent without imposing the assumption on the amortization network. To see this, we randomly generate fifty tasks from a dataset, where classes may appear a number of times in different tasks. We then perform free-form (non-amortized) variational inference on the weights for each of the tasks, with a Gaussian variational distribution:

$$q_\phi\left(W^{(t)}|\mathcal{D}^{(t)}, \theta\right) = \mathcal{N}\left(W^{(t)}; \mu_\phi^{(t)}, \sigma_\phi^{(t)2}\right). \tag{B.3}$$

If the assumption is reasonable, we may expect the distribution of the weights of a specific class to be similar regardless of the additional classes in the task.

We examine 5-way classification in the MNIST dataset. We randomly sample and fix fifty such tasks. We train the model twice using the same feature extraction network used in the few-shot classification experiments, and fix the $d_\theta$ to be 16 and 2. We then train the model in an episodic manner by minibatching tasks at each iteration. The model is trained to convergence, and achieves 99% accuracy on held out test examples for the tasks. After training is complete we examine the optimized $\mu_\phi^{(t)}$ for each class in each task. Fig. B.1a shows a t-SNE (Maaten and Hinton, 2008) plot for the 16-dimensional weights. We see that when reduced to 2-dimensions, the weights cluster according to class. Fig. B.1b visualizes the weights in their original space. In this plot, weights from the same class are grouped together, and clear similarity patterns are evident across the image, showing that weights from the same class have similar means across tasks. Fig. B.2 details the task weights in 2-dimensional space. Here, each pentagon represents the weight means learned for one training task, where the nodes of the pentagon are colored according to the class the weights represent. In Fig. B.2a we see that overall,

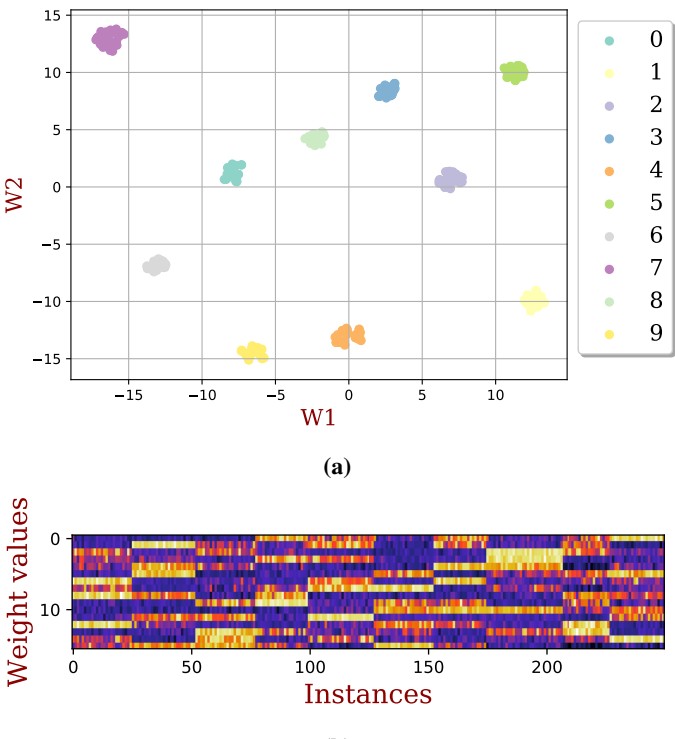

**(a)**

**(b)**

**Figure B.1:** Visualizing the learned weights for $d_\theta = 16$. (a) Weight dimensionality is reduced using T-SNE (Maaten and Hinton, 2008). Weights are colored according to class. (b) Each weight represents one column of the image. Weights are grouped by class.

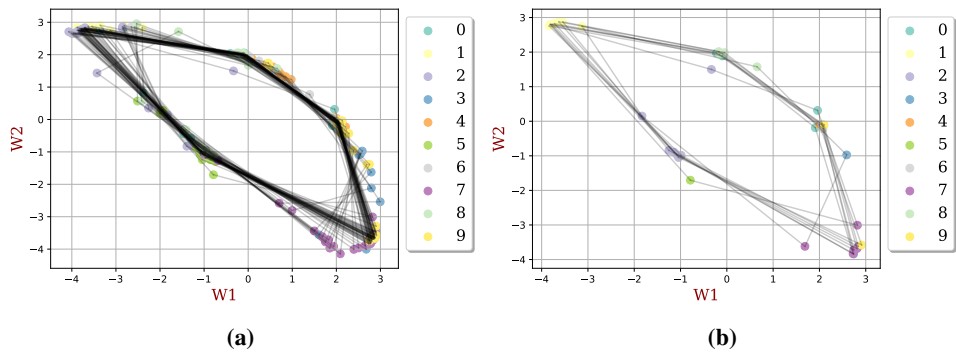

**(a)**                                       **(b)**

**Figure B.2:** Visualizing the task weights for $d_\theta = 2$. (a) All training tasks. (b) Only training tasks containing both '1's and '2's.

the classes cluster in 2-dimensional space as well. However, there is some overlap (e.g., classes '1' and '2'), and that for some tasks a class-weight may appear away from the cluster. Fig. B.2b shows the same plot, but only for tasks that contain both class '1' and '2'. Here we can see that for these tasks, class '2' weights are all located away from their cluster.

This implies that each class-weights are typically well-approximated as being independent of the task. However, if the model lacks capacity to properly assign each set of class weights to different regions of space, for tasks where classes from similar regions of space appear, the inference procedure will 'move' one of the class weights to an 'empty' region of the space.

## C  VARIATIONAL INFERENCE DERIVATIONS FOR THE MODEL

We derive a VI-based objective for our probabilistic model. By "amortized" VI we mean that $q_\phi(\psi|D^{(t)}, \theta)$ is parameterized by a neural network with a fixed-sized $\phi$. Conversely, "non-amortized" VI refers to local parameters $\phi^{(t)}$ that are optimized independently (at test time) for each new task $t$, such that $q(\psi|D^{(t)}, \theta) = \mathcal{N}(\psi|\mu_{\phi^{(t)}}, \Sigma_{\phi^{(t)}})$. However, the derivation of the objective function does not change between these options. For a single task $t$, an evidence lower bound (ELBO; (Wainwright and Jordan, 2008)) may be expressed as:

$$\mathcal{L}_t = \mathbb{E}_{q_\phi(\psi|D^{(t)}, \theta)} \left[ \sum_{(x,y) \in D^{(t)}} \log p(y|x, \psi, \theta) \right] - \mathrm{KL} \left[ q_\phi(\psi|D^{(t)}, \theta) \| p(\psi|\theta) \right]. \tag{C.1}$$

We can then derive a stochastic estimator to optimize Eq. (C.1) by sampling $D^{(t)} \sim p(D)$ (approximated with a training set of tasks) and simple Monte Carlo integration over $\psi$ such that $\psi^{(l)} \sim q_\phi(\psi|D^{(t)}, \theta)$:

$$\hat{\mathcal{L}} = \frac{1}{T} \sum_{t=1}^{T} \left( \sum_{(x,y) \in D^{(t)}} \left( \frac{1}{L} \sum_{l=1}^{L} \log p(y|x, \psi^{(l)}, \theta) \right) - \mathrm{KL} \left[ q_\phi(\psi|D^{(t)}, \theta) \| p(\psi|\theta) \right] \right), \tag{C.2}$$

Eq. (C.2) differs from our objective function in Eq. (4) in two important ways: (i) Eq. (4) does not contain a KL term for $q_\phi(\psi|D^{(t)}, \theta)$ (nor any other form of prior distribution over $\psi$, and (ii) Eq. (C.1) does not distinguish between training and test data within a task, and therefore does not explicitly encourage the model to generalize in any way.

## D  EXPERIMENTATION DETAILS

In this section we provide comprehensive details on the few-shot classification experiments.

### D.1  OMNIGLOT FEW-SHOT CLASSIFICATION TRAINING PROCEDURE

Omniglot (Lake et al., 2011) is a few-shot learning dataset consisting of 1623 handwritten characters (each with 20 instances) derived from 50 alphabets. We follow a pre-processing and training procedure akin to that defined in (Vinyals et al., 2016). First the images are resized to $28 \times 28$ pixels and then character classes are augmented with rotations of 90 degrees. The training, validation and test sets consist of a random split of 1100, 100, and 423 characters, respectively. When augmented this results in 4400 training, 400 validation, and 1292 test classes, each having 20 character instances. For $C$-way, $k_c$-shot classification, training proceeds in an episodic manner. Each training iteration consists of a batch of one or more tasks. For each task $C$ classes are selected at random from the training set. During training, $k_c$ character instances are used as training inputs and 15 character instances are used as test inputs. The validation set is used to monitor the progress of learning and to select the best model to test, but does not affect the training process. Final evaluation of the trained model is done on 600 randomly selected tasks from the test set. During evaluation, $k_c$ character instances are used as training inputs and $k_c$ character instances are used as test inputs. We use the Adam (Kingma and Ba, 2015) optimizer with a constant learning rate of 0.0001 with 16 tasks per batch to train all models. The 5-way - 5-shot and 5-way - 1-shot models are trained for 80,000 iterations while the 20-way - 5-shot model is trained for 60,000 iterations, and the 20-way - 1-shot model is trained for 100,000 iterations. In addition, we use a Gaussian form for $q$ and set the number of $\psi$ samples to $L = 10$.

## D.2 *mini*IMAGENET FEW-SHOT CLASSIFICATION TRAINING PROCEDURE

*mini*ImageNet (Vinyals et al., 2016) is a dataset of 60,000 color images that is sub-divided into 100 classes, each with 600 instances. The images have dimensions of $84 \times 84$ pixels. For our experiments, we use the 64 training, 16 validation, and 20 test class splits defined by (Ravi and Larochelle, 2017). Training proceeds in the same episodic manner as with Omniglot. We use the Adam (Kingma and Ba, 2015) optimizer and a Gaussian form for $q$ and set the number of $\psi$ samples to $L = 10$. For the 5-way - 5-shot model, we train using 4 tasks per batch for 100,000 iterations and use a constant learning rate of 0.0001. For the 5-way - 1-shot model, we train with 8 tasks per batch for 50,000 iterations and use a constant learning rate of 0.00025.

## D.3 FEW-SHOT CLASSIFICATION NETWORK ARCHITECTURES

Tables D.1 to D.4 detail the neural network architectures for the feature extractor $\theta$, amortization network $\phi$, and linear classifier $\psi$, respectively. The feature extraction network is very similar to that used in (Vinyals et al., 2016). The output of the amortization network yields mean-field Gaussian parameters for the weight distributions of the linear classifier $\psi$. When sampling from the weight distributions, we employ the local-reparameterization trick (Kingma et al., 2015), that is we sample from the implied distribution over the logits rather than directly from the variational distribution. To reduce the number of learned parameters, we share the feature extraction network $\theta$ with the pre-processing phase of the amortizaion network $\psi$.

**Table D.1:** Feature extraction network used for Omniglot few-shot learning. Batch Normalization and dropout with a keep probability of 0.9 used throughout.

**Omniglot Shared Feature Extraction Network ($\theta$):** $\tilde{x} \to h_\theta(\tilde{x})$

| Output size | Layers |
|---|---|
| $28 \times 28 \times 1$ | Input image |
| $14 \times 14 \times 64$ | conv2d ($3 \times 3$, stride 1, SAME, RELU), dropout, pool ($2 \times 2$, stride 2, SAME) |
| $7 \times 7 \times 64$ | conv2d ($3 \times 3$, stride 1, SAME, RELU), dropout, pool ($2 \times 2$, stride 2, SAME) |
| $4 \times 4 \times 64$ | conv2d ($3 \times 3$, stride 1, SAME, RELU), dropout, pool ($2 \times 2$, stride 2, SAME) |
| $2 \times 2 \times 64$ | conv2d ($3 \times 3$, stride 1, SAME, RELU), dropout, pool ($2 \times 2$, stride 2, SAME) |
| 256 | flatten |

**Table D.2:** Feature extraction network used for *mini*ImageNet few-shot learning. Batch Normalization and dropout with a keep probability of 0.5 used throughout.

***mini*ImageNet Shared Feature Extraction Network ($\theta$):** $\tilde{x} \to h_\theta(\tilde{x})$

| Output size | Layers |
|---|---|
| $84 \times 84 \times 1$ | Input image |
| $42 \times 42 \times 64$ | conv2d ($3 \times 3$, stride 1, SAME, RELU), dropout, pool ($2 \times 2$, stride 2, VALID) |
| $21 \times 21 \times 64$ | conv2d ($3 \times 3$, stride 1, SAME, RELU), dropout, pool ($2 \times 2$, stride 2, VALID) |
| $10 \times 10 \times 64$ | conv2d ($3 \times 3$, stride 1, SAME, RELU), dropout, pool ($2 \times 2$, stride 2, VALID) |
| $5 \times 5 \times 64$ | conv2d ($3 \times 3$, stride 1, SAME, RELU), dropout, pool ($2 \times 2$, stride 2, VALID) |
| $2 \times 2 \times 64$ | conv2d ($3 \times 3$, stride 1, SAME, RELU), dropout, pool ($2 \times 2$, stride 2, VALID) |
| 256 | flatten |

# E SHAPENET EXPERIMENTATION DETAILS

## E.1 VIEW RECONSTRUCTION TRAINING PROCEDURE AND NETWORK ARCHITECTURES

ShapeNetCore v2 (Chang et al., 2015) is an annotated database of 3D objects covering 55 common object categories with $\sim$51,300 unique objects. For our experiments, we use 12 of the largest object categories. Refer to Table E.1 for a complete list. We concatenate all instances from all 12 of the

**Table D.3:** Amortization network used for Omniglot and *mini*ImageNet few-shot learning.

**Amortization Network** $(\phi)$: $x_1^c, ..., x_{k_c}^c \rightarrow \mu_{w^{(c)}}, \sigma^2_{w^{(c)}}$

| Phase | Output size | Layers |
|---|---|---|
| feature extraction | $k \times 256$ | shared feature network $(\theta)$ |
| instance pooling | 256 | mean |
| $\psi$ weight distribution | 256 | $2 \times$ fully connected, ELU + |
| | | linear fully connected to $\mu_{w^{(c)}}, \sigma^2_{w^{(c)}}$ |

**Table D.4:** Linear classifier used for Omniglot and *mini*ImageNet few-shot learning.

**Linear Classifier** $(\psi)$: $h_\theta(\tilde{x}) \rightarrow p(\tilde{y}|\tilde{x}, \theta, \psi_t)$

| Output size | Layers |
|---|---|
| 256 | Input features |
| $C$ | fully connected, softmax |

object categories together to obtain a dataset of 37,108 objects. This concatenated dataset is then randomly shuffled and we use 70% of the objects (25,975 in total) for training, 10% for validation (3,710 in total) , and 20% (7423 in total) for testing. For each object, we generate $V = 36$, $128 \times 128$ pixel image views spaced evenly every 10 degrees in azimuth around the object. We then convert the rendered images to gray-scale and reduce their size to be $32 \times 32$ pixels. Again, we train our model in an episodic manner. Each training iteration consists a batch of one or more tasks. For each task an object is selected at random from the training set. We train on a single view selected at random from the $V = 36$ views associated with each object and use the remaining 35 views to evaluate the objective function. We then generate 36 views of the object with a modified version of our amortization network which is shown diagrammatically in Fig. 3. To evaluate the system, we generate views and compute quantitative metrics over the entire test set. Tables E.2 to E.4 describe the network architectures for the encoder, amortization, and generator networks, respectively. To train, we use the Adam (Kingma and Ba, 2015) optimizer with a constant learning rate of 0.0001 with 24 tasks per batch for 500,000 training iterations. In addition, we set $d_\phi = 256$, $d_\psi = 256$ and number of $\psi$ samples to 1.

**Table E.1:** List of ShapeNet categories used in the VERSA view reconstruction experiments.

| Object Category | sysnet ID | Instances |
|---|---|---|
| airplane | 02691156 | 4045 |
| bench | 02828884 | 1813 |
| cabinet | 02933112 | 1571 |
| car | 02958343 | 3533 |
| phone | 02992529 | 831 |
| chair | 03001627 | 6778 |
| display | 03211117 | 1093 |
| lamp | 03636649 | 2318 |
| speaker | 03691459 | 1597 |
| sofa | 04256520 | 3173 |
| table | 04379243 | 8436 |
| boat | 04530566 | 1939 |

**Table E.2:** Encoder network used for ShapeNet few-shot learning. No dropout or batch normalization is used.

**ShapeNet Encoder Network ($\phi$):** $y \to h$

| Output size | Layers |
|---|---|
| $32 \times 32 \times 1$ | Input image |
| $16 \times 16 \times 64$ | conv2d ($3 \times 3$, stride 1, SAME, RELU), pool ($2 \times 2$, stride 2, VALID) |
| $8 \times 8 \times 64$ | conv2d ($3 \times 3$, stride 1, SAME, RELU), pool ($2 \times 2$, stride 2, VALID) |
| $4 \times 4 \times 64$ | conv2d ($3 \times 3$, stride 1, SAME, RELU), pool ($2 \times 2$, stride 2, VALID) |
| $2 \times 2 \times 64$ | conv2d ($3 \times 3$, stride 1, SAME, RELU), pool ($2 \times 2$, stride 2, VALID) |
| $d_\phi$ | fully connected, RELU |

**Table E.3:** Amortization network used for ShapeNet few-shot learning.

**ShapeNet Amortization Network ($\phi$):** $x_1^{(t)}, ..., x_k^{(t)}, y_1^{(t)}, ..., y_k^{(t)} \to \mu_\psi, \sigma_\psi^2$

| Phase | Output size | Layers |
|---|---|---|
| $\phi_{pre}$ | $k \times d_\phi$ | encoder network ($\phi$) |
| concatenate $h$ and $X$ | $k \times (d_\psi + d_X)$ | concat($h$, $X$) |
| $\phi_{mid}$ | $k \times d_\phi$ | $2 \times 2$ fully connected, ELU |
| instance pooling | $1 \times d_\phi$ | average |
| $\phi_{post}$ | $1 \times d_\phi$ | $2\times$ fully connected, ELU |
| $\psi$ distribution | $d_\psi$ | fully connected linear layers to $\mu_\psi, \sigma_\psi^2$ |

**Table E.4:** Generator network used for ShapeNet few-shot learning. No dropout or batch normalization is used.

**ShapeNet Generator Network ($\theta$):** $\tilde{x} \to p(\tilde{y}|\tilde{x}, \theta, \psi^{(t)})$

| Output size | Layers |
|---|---|
| $d_\psi + d_x$ | concat($\psi$, $x$) |
| 512 | fully connected, RELU |
| 1024 | fully connected, RELU |
| $2 \times 2 \times 256$ | reshape |
| $4 \times 4 \times 128$ | deconv2d ($3 \times 3$, stride 2, SAME, RELU) |
| $8 \times 8 \times 64$ | deconv2d ($3 \times 3$, stride 2, SAME, RELU) |
| $16 \times 16 \times 32$ | deconv2d ($3 \times 3$, stride 2, SAME, RELU) |
| $32 \times 32 \times 1$ | deconv2d ($3 \times 3$, stride 2, SAME, sigmoid) |

