# OpenReview forum: "Meta-Learning Probabilistic Inference for Prediction"
_ICLR.cc/2019/Conference_

### Official Review · AnonReviewer1 · 2018-10-24
**Few-shot learning, based on amortized inference network for parameters of logistic regression head models. Uses learning criterion based on predictive distributions on train/test splits. Extensive comparison, achieves state-of-the-art despite simpler setup than many competitors**

**Rating:** 8
**Confidence:** 4

**Review:**

Summary:
This work tackles few-shot (or meta) learning from a probabilistic inference viewpoint. Compared to previous work, it uses a simpler setup, performing task-specific inference only for single-layer head models, and employs an objective based on predictive distributions on train/test splits for each task (rather than an approximation to log marginal likelihood). Inference is done amortized by a network, whose input is the task training split. The same network is used for parameters of each class (only feeding training points of that class), which allows an arbitrary number of classes per task. At test time, inference just requires forward passes through this network, attractive compared to non-amortized approaches which need optimization or gradients here.

It provides a clean, decision-theoretic derivation, and clarifies relationships to previous work. The experimental results are encouraging: the method achieves a new best on 5-way, 5-shot miniImageNet, despite the simple setup. In general, explanations in the main text could be more complete (see questions). I'd recommend shortening Section 4, which is pretty obvious.

- Quality: Several interesting differences to prior work. Well-done experiments
- Clarity: Clean derivation, easy to understand. Some details could be spelled out better
- Originality: Several important novelties (predictive criterion, simple model setup, amortized inference network). Closely related to "neural processes" work, but this happened roughly at the same time
- Significance: The few-shot learning results are competitive, in particular given they use a simpler model setup than most previous work. I am not an expert on these kind of experiments, but I found the comparisons fair and rather extensive

Interesting about this work:
- Clean Bayesian decision-theoretic viewpoint. Key question is of course whether
   an inference network of this simple structure (no correlations, sum combination
   of datapoints, same network for each class) can deliver a good approximation to
   the true posterior.
- Different to previous work, task-specific inference is done only on the weights of
   single-layer head models (logistic regression models, with shared features).
   Highly encouraging that this is sufficient for state-of-the-art few-shot classification
   performance. The authors could be more clear about this point.
- Simple and efficient amortized inference model, which along with the neural
   network features, is learned on all data jointly
- Optimization criterion is based on predictive distributions on train/test splits, not
   on the log marginal likelihood. Has some odd consequences (question below),
   but clearly works better for few-shot classification

Experiments:
- 5.1: Convincing results, in particular given the simplicity of the model setup and
   the inference network. But some important points are not explained:
   - Which of the competitors (if any) use the same restricted model setup (inference
      only on the top-layer weights)? Clearly, MAML does not, right? Please state this
      explicitly.
   - For Versa, you use k_c training and 15 test points per task update during
      training. Do competitors without train/test split also get k_c + 15 points, or
      only k_c points? The former would be fair, the latter not so much.
- 5.2: This seems a challenging problem, and both your numbers and reconstructions
   look better than the competitor. I cannot say more, based on the very brief
   explanations provided here.
   The main paper does not really state what the model or the likelihood is. From
   F.4 in the Appendix, this model does not have the form of your classification
   models, but psi is input at the bottom of the network. Also, the final layer has
   sigmoid activation. What likelihood do you use?
   One observation: If you used the same "inference on final layer weights" setup
   here, and Gaussian likelihood, you could compute the posterior over psi in closed
   form, no amortization needed. Would this setup apply to your problem?

Further questions:
- Confused about the input to the inference network. Real Bayesian inference would
   just see features h_theta(x) as inputs, not the x's. Why not simply feed features in
   then?
   Please do improve the description of the inference network, this is a major
   novelty of this paper, and even the appendix is only understandable by reading
   other work as well. Be clear how it depends on theta (I think nothing is lost by
   feeding in the h_theta(x)).
- The learning criterion based on predictive distributions on train/test splits seem
   to work better than ELBO-like criteria, for few-shot classification.
   But there are some worrying aspects. The marginal likelihood has an Occam's
   razor argument to prevent overfitting. Why would your criterion prevent overfitting?
   And it is quite worrying that the prior p(psi | theta) drops out of the method
   entirely. Can you comment more on that?

Small:
- p(psi_t | tilde{x}_t, D_t, theta) should be p(psi_t | D_t, theta). Please avoid a more
   general notation early on, if you do not do it later on. This is confusing

---

> ### Author Response · Authors · 2018-11-13
> **Responding to your review**
>
> “Which of the competitors (if any) use the same restricted model setup (inference only on the top-layer weights)?”
>
> To the best of our knowledge, almost all the competing methods adapt the entire network for new tasks. We have amended the paper to clarify this point (see Section 5.2).
>
> “Do competitors without train/test split also get k_c + 15 points, or only k_c points?”
>
> To the best of our knowledge, all methods we compare to use train/test splits, with the exception of the VI methods referenced in Table 1. The VI methods used the same number of observations at train time (i.e., the data available to all methods was identical).
>
> “The main paper does not really state what the model or the likelihood is [in the ShapeNet experiments]. From F.4 in the Appendix, this model does not have the form of your classification models, but psi is input at the bottom of the network. Also, the final layer has sigmoid activation. What likelihood do you use?”
>
> The terseness of the ShapeNet model details was a result of space constraints. We have amended the paper to include additional explanatory details (see Section 3). You are correct in observing that psi plays a different role from the classification case, namely as an input to the image-generator. The likelihood we used is Gaussian, the sigmoid activation ensures that the mean is between 0 and 1, reflecting the constraints on pixel-intensities. Your observation that using top-layer weights would allow us to perform exact inference is very insightful. We decided to use an architecture that passed the latent parameters underlying each shape instance through multiple non-linearities, but it would be very interesting to compare to the simpler baseline that you suggest. As this is a significant undertaking, we will leave it to future work,
>
> “Real Bayesian inference would just see features h_theta(x) as inputs, not the x's. Why not simply feed features in then? … Be clear how it depends on theta (I think nothing is lost by feeding in the h_theta(x)).”
>
> Thank you for suggesting this cleaner way of presenting our work. We agree with your observations on the input to the inference network. We have amended Fig. 2 accordingly, and have improved the descriptions in Section 3.
>
> “The marginal likelihood has an Occam's razor argument to prevent overfitting. Why would your criterion prevent overfitting?”
>
> The mechanism preventing overfitting in our criterion is the meta train / test splits, which explicitly encourages the model to generalize from the training observations to the test data. Methods based on held-out sets, like cross validation, are known to favor models which are more complex than those favoured by Bayesian model comparison [i, ii]. However, as is empirically demonstrated in the experimental section, our proposed criterion consistently outperformed variational objectives.
>
> “It is quite worrying that the prior p(psi | theta) drops out of the method entirely. Can you comment more on that?”
>
> This is a subtle point that we view as both a feature and a bug. It is a feature in the sense that a prior is learned implicitly through the sampling procedure (as is shown for example in the simple Gaussian experiment -- see Section 5.1). This can be compared to VI, for example, where the prior enters through a KL regularization term which often favours underfitting. It is a bug if, for example, the user has a priori knowledge about the parameters that they would like to leverage. In this case, it could be possible to use synthetic training data to incorporate such knowledge into the scheme.  However, for the predictive purposes explored in this work, we did not find that the lack of prior posed an issue.
>
>
> [i] - C. E. Rasumessen and Z. Ghahramani. Occam’s razor. 2001.
> [ii] - I. Murray and Z. Ghahramani. A note on the evidence and Bayesian Occam’s razor. 2005.

---

### Official Review · AnonReviewer3 · 2018-11-01
**Review for Meta-Learning Probabilistic Inference for Prediction**

**Rating:** 6
**Confidence:** 2

**Review:**

This paper presents two different sections:
1. A generalized framework to describe a range of meta-learning algorithms.
2. A meta-learning algorithm that allows few shot inference over new tasks without the need for retraining. The important aspect of the algorithm is the context independence assumption between posteriors of different classes for learning weights. This reduces the number of parameters to amortize during meta-training. More importantly, it makes it independent of the number of classes in a task, and effectively doing meta-training across class inference instead of each task. The idea sounds great, but I am skeptical of the justification behind the independence assumption which, as per its justifications sounds contrived and only empirical.

Overall, I feel the paper makes some progress in important aspects of meta-learning.

---

> ### Author Response · Authors · 2018-11-13
> **On justification for the context-indepence assumption**
>
> “The important aspect of the algorithm is the context independence assumption between posteriors of different classes for learning weights. … The idea sounds great, but I am skeptical of the justification behind the independence assumption which, as per its justifications sounds contrived and only empirical.”
>
> We thank the reviewer for imploring us to think more carefully about this point. We share the concern that providing only an empirical justification for the context independent assumption is slightly troubling. We have therefore considered this more carefully, and have found that there is a principled justification of this design choice, which is best understood through the lens of density ratio estimation [i, ii].
>
> Results from Density Ratio Estimation [i, ii] show that an optimal softmax classifier learns the ratio of the densities
>
>     Softmax(y=k | x) = p(x | y=k) / Sum_j p(x | y=j)
>
> assuming equal a priori probability for each class. Our system follows his optimal form by setting:
>
>             log p(\tilde{x} | y=c) proportional h_theta ( \tilde{x})^T w_c
>
> where w_c ~ q_phi (w | {x_n ; y_n=c} ) for each class in a given task. Here {(x_n, y_n)} are the few-shot training examples, and $\tilde{x}$ is the test example. This argument states that under ideal conditions (i.e., we can perfectly estimate p(y=c | x) ), the context-independent assumption is correct, and further motivates our design.
>
> We have amended the paper to include this argument (see Appendix B). We thank the reviewer for pointing to this important issue, and we hope that this alleviates some of their concerns.
>
> [i] - S. Mohamed. The Density Ratio Trick. The Spectator (Blog). 2018
> [ii] - M. Sugiyama, T. Suzuki, and T. Kanamori. Density ratio estimation in machine learning. 2012

---

### Official Review · AnonReviewer2 · 2018-11-03
**A novel meta-learning framework**

**Rating:** 7
**Confidence:** 4

**Review:**

This paper proposes both a general meta-learning framework with approximate probabilistic inference, and implements an instance of it for few-shot learning. First, they propose Meta-Learning Probabilistic inference for Prediction (ML-PIP) which trains the meta-learner to minimize the KL-divergence between the approximate predictive distribution generated from it and predictive distribution for each class. Then, they use this framework to implement Versatile Amortized Inference, which they call VERSA. VERSA replaces the optimization for test time with efficient posterior inference, by generating distribution over task-specific parameters in a single forward pass. The authors validate VERSA against amortized and non-amortized variational inference which it outperforms. VERSA is also highly versatile as it can be trained with varying number of classes and shots.

Pros
- The proposed general meta-learning framework that aims to learn the meta-learner that approximates the predictive distribution over multiple tasks is quite novel and makes sense.
- VERSA obtains impressive performance on both benchmark datasets for few-shot learning and is versatile in terms of number of classes and shots.
- The appendix section has in-depth analysis and additional experimental results which are quite helpful in understanding the paper.

Cons
- The main paper feels quite empty, especially the experimental validation parts with limited number of baselines. It would have been good if some of the experiments could be moved into the main paper. Some experimental results such as Figure 4 on versatility does not add much insight to the main story and could be moved to appendix.
- It would have been good if there was some validation of the time-performance of the model as one motivation of meta-learning is rapid adaptation to a test-time task.

In sum, since the proposed meta-learning probabilistic inference framework is novel and effective I vote for accepting the paper. However the structure and organization of the paper could be improved by moving some of the methodological details and experimental results in the appendix to the main paper.

---

> ### Author Response · Authors · 2018-11-13
> **More experiments to main text + timing experiments**
>
> “It would have been good if some of the experiments could be moved into the main paper. … the structure and organization of the paper could be improved by moving some of the methodological details and experimental results in the appendix to the main paper.”
>
> We agree that a significant portion of interesting content has been relegated to the appendix in our submission. Much of this, of course, has to do with space constraints. However, we have addressed this in the revised version in line with your suggestions by (i) moving the appendix containing the toy-data experimentation to the main body of the paper (see Section 5.1), and (ii) moving some methodological details from the appendix in to the experiments section (see Section 5).
>
> “It would have been good if there was some validation of the time-performance of the model as one motivation of meta-learning is rapid adaptation to a test-time task. “
>
> We strongly agree that the issue of performance timing is of great interest, and it is useful and important to validate this experimentally. We were originally hesitant to add any timing results as code released with research papers is often optimized for correctness as opposed to speed. That said, we measured the test time performance of both MAML (as implemented in the authors'  publicly available repository at https://github.com/cbfinn/maml) and Versa in 5-shot 5-way experiments on mini-ImageNet, using the same architectures for both. We found Versa to achieve 5x speed up compared to MAML, while achieving significantly better accuracy (see Table 3). We have amended the paper to include this experimental data (see Section 5.2 for details). We believe this data demonstrates the performance gains achieved by relieving the need for test time optimization procedures.

---

### Author Response · Authors · 2018-11-13
**Thank you for your reviews and comments**

Dear Reviewers,

Many thanks for your detailed comments and suggestions. We really appreciate the time and effort you have put into reading our paper. Your comments are both insightful and constructive, and we believe have contributed to improving the quality of our paper.

We have uploaded a revised version of the paper, incorporating your comments and suggestions. Below, we address each of your reviews individually.

---

### Public Comment · (anonymous) · 2018-12-10
**Incorrect results of Prototypical Nets in Table 3?**

Could you please comment on why the results of Prototypical Nets mentioned in Table 3 of your submission are different (lower) from those reported by the authors of the model in their paper [1]? Especially this concerns the 5-shot 5-way miniImageNet setup where their result is 68,20 +/- 0.66% which contradicts your claim of getting state-of-the-art results on any of the standard few-shot learning benchmarks. This is especially strange since the arxiv version of your paper includes the correct numbers. Thank you in advance for your answer.

[1] - Snell, Jake, Kevin Swersky, and Richard Zemel. Prototypical networks for few-shot learning. Advances in Neural Information Processing Systems. 2017.

---

> ### Author Response · Authors · 2018-12-11
> **Prototypical Nets propose several experimental protocols**
>
> Thank you for your question!
>
> The prototypical networks paper proposes a number of different experimental protocols. One of these protocols trains on higher "way" than what is ultimately used for testing. This is detailed in table 5 in Appendix B of the prototypical networks paper. There you will see that the number you are quoting is achieved by training the system to perform 20 way classification, and then testing it on 5 way classification (final row of the table). This experimental protocol differs from that used by all other methods. When matching experimental protocols are used, i.e., training and testing on 5-way classification, prototypical networks achieve the numbers we quote in our ICLR submission (row 10 of their table).
>
> The discrepancy between the experimental protocols was pointed out to us by readers of the arxiv version of our paper who suggested reporting numbers from the same experimental protocol. Hence the differing numbers between this submission and the arxiv version. You are correct in pointing out that in the unconstrained setting prototypical networks achieve better performance, but we are of the opinion that a fairer comparison is made when all methods use the same experimental protocol. The note in our submission saying "The tables include results for only those approaches with comparable training procedures ..." was intended to clarify this, but we will clarify this more explicitly in the final version if it is accepted.
>
> Finally, note that even the improved prototypical networks number (68.20 ± 0.66%) is within error bars of the Versa number (67.37 ± 0.86%).

---

> > ### Public Comment · (anonymous) · 2018-12-12
> > **Question regarding your protocol: is your current comparison really fair?**
> >
> > In your submission it is stated that during training you used 16 tasks per batch for all four Omniglot setups, 4 tasks per batch for 5-shot 5-way task on miniImageNet and 8 tasks per batch for 1-shot 5-way task on miniImageNet. Technically, if we take a look at the features right before the softmax, due to the context independence assumption between the posteriors of different classes there is no difference between running four parallel 5-way tasks and one 20-way task, as long as you don't update the weights while running these four tasks. So your learning procedure can be viewed as a 20-way classification with masked softmax. From this point of view it seems that the Prototypical Networks try to learn a more complicated 20-way task during the training stage while your model is trained on a "simplified and masked" 5-way task while having access to the same 20 classes (in the 5-way 5-shot miniImagenet setup, and it has access to a much larger number of classes and total number of query samples in all the other setups, more query samples that the Prototypical Networks have access to during a single update of the fully shared model). The difference is only in the losses, and it is okay (I guess?) to have different losses (your way of computing losses can be viewed as a particular, masked instance of their way). It seems that if we are talking about truly fair comparison to the results by Prototypical Networks that you mention in your current version of the paper, your model should be trained with a single task per meta-batch, at least so that the number of query samples used for a single model update was comparable (now it is 15 * 5 for Prototypical Networks and 15 * 20 for 5-way 5-shot and 15 * 40 for 5-way 1-shot VERSA models on miniImageNet), otherwise it is also not fair.
> >
> > Finally, there is a difference between the claims "within the error bars for all standard benchmarks" and "sets new SOTA results by a certain margin over the pervious best".

---

> > > ### Author Response · Authors · 2018-12-12
> > > **Yes, the comparison is fair for the following reasons.**
> > >
> > > Thanks for the follow up question!
> > >
> > > Existing comparisons:
> > > - As is shown in Section 4, prototypical networks is a special case of Versa, allowing for a direct comparison of the two when the training procedure is equivalent.
> > > - Our training procedure (and that of other methods presented in Table 3) is equivalent to the "matched training condition" in prototypical nets (i.e., train and test "way" are the same).
> > > - Experimental protocols (and in particular, makeup of the mini-batches) were found not to have a substantial effect on the performance of Versa. The experimental protocols described in the paper (number of tasks per batch, size of test sets, etc') closely follow that of MAML. See below for more details.
> > > - For example, we ran Versa with a meta-training batch of 1 (directly equivalent to the prototypical network setup), and found the final accuracy is within error-bars of the result of the submission, and still above the errors bars of prototypical networks trained and tested on 5-way.
> > > - We therefore consider the comparisons presented to be direct and fair.
> > >
> > > Comparisons using higher way training
> > > - We have not investigated the effects of training with a higher way than testing.
> > > - This changes the objective function due to the normalisation constant in the softmax e.g. this would have a single normaliser for all 20 classes if considered together, versus separate ones for each of the tasks if the classes were split into 4 tasks of 5 classes each.
> > > - This is the key difference between these two training conditions and is not something Versa currently exploits.
> > > - We agree that it would be interesting to test whether using this modified objective improved Versa and indeed whether the same idea could lead to improvements in other methods too.
> > >
> > > More details on the training protocol
> > > - For both Omniglot and miniImageNet, experiments demonstrated that the performance of Versa on the meta-test set is not sensitive to the number of tasks per batch during training.
> > > - As such, the experimental protocol (number of tasks per batch and the number of query points per batch) for both miniImageNet and Omniglot were chosen to match the MAML protocol.
> > > - We also ran the experiments with meta-training batch of size 1, which is directly equivalent to the prototypical network setup.
> > > - The performance in these experiments was very similar to our best results (within error bars), and significantly better than what is reported by prototypical nets for the same setup.
> > > - In summary the final performance was not found to be particularly sensitive to these choices.
> > > - Arguably, this is to be expected as this is equivalent to selecting the size of the mini-batch in conventional learning. In particular, our model makes an independence assumption across tasks (given \theta).
> > > - As prototypical networks are a special case of Versa (with the amortization network set to identity around the mean encoding), we expect similar findings to hold for this model.

---

> > > > ### Public Comment · (anonymous) · 2018-12-12
> > > > **I'm still not convinced at all. Matter of wordings and formalities.**
> > > >
> > > > Thank you for your detailed replies.
> > > >
> > > > First of all, please, provide the numbers. There are a lot of phrases like "within error bars", "no substantial effect". Please, be specific. While your model with 1 task per meta-batch might be within the error bars of your current submission, it might no longer be within the error-bars of current SOTA results on some of the benchmarks.
> > > >
> > > > Matching to MAML protocol is understandable, however, the Prototypical Networks are trained using a single episode (=single task per meta-batch) per shared model update. If you insist on your current training protocol, the Prototypical Networks need to be trained using the same number of episodes per single task-specific (in fact shared!) update as you use. You need to truly match Prototypical Networks protocol to the one you and MAML use. Otherwise report your numbers with a single task per meta-batch protocol, whether they are within the error-bars of your current submission or not.
> > > >
> > > > It certainly makes sense to take into consideration the number of query samples used for a single task-specific model (in fact HEAVILY shared across different tasks). The more query samples you use, the more regularised is the training of the shared feature extractor that both VERSA and Prototypical Nets utilise. You use from 15 * 20 to 15 * 40 samples for miniImageNet while the Prototypical Nets from your table only use 15 * 5. This is the advantage you give yourself by comparing your multitask-per-meta-batch protocol to the single-task-per-meta-batch protocol in Prototypical Networks.
> > > >
> > > > In addition to that, if we are talking about using truly comparable settings for all models, similarly to MAML and the Prototypical Nets, your feature extractor should consist of 4 convolutional blocks for miniImageNet dataset, not 5, and the number of features better also be similar.  I understand that this might not change the relative positions, however, it should be interesting to see what would be the margin between the results in this case. This would help to evaluate the advantage of using additional network compared to the Prototypical Nets.
> > > >
> > > > Another approach would be to ignore all those minor differences in the setups and objectives and use acknowledged peer-reviewed best results for all models and try to make your model as good as possible. While it is good to have an additional table that compares the models in the almost exact settings, different models might benefit from different tricks, so these tricks should not be ignored in final assessment of the model. Absence of the best result by the Prototypical Networks together with your claims about setting SOTA is certainly misleading. It is your job to demonstrate the best potential of your model, so there needs to be a comparison to the Prototypical Networks at their best, even if it requires using additional protocol (even though I still consider your protocol to be interpretable within the protocol of the Prototypical Nets, you repeated my explanation with your own words confirming the the difference is only in the objective). The lack of a valid comparison is certainly a drawback.
> > > >
> > > > If one discovers they can use some valid trick to improve the results of their model, they should not avoid using it. The Prototypical Networks are learning a more complicated task during the training stage than any of the other models in the table while using the same amount of data, and they should not be punished for that. It is not that the size of their model was drastically different (in fact, it is the smallest since they only use the same feature extractor as you and MAML do) or that the quality of their model was increasing linearly with the larger number of the classes used during the training. There is a trade-off, they explored it and used for regularising their model, in the same way other people use different architectures or objectives.

---

> > > > > ### Author Response · Authors · 2018-12-13
> > > > > **Numbers using prototypical networks matched training protocol**
> > > > >
> > > > > Thanks for the follow up questions.
> > > > >
> > > > > The experiments with 1 task per batch yield the following result on 5-way 5-shot learning with Versa: (66.75 + / - 0.9)%. This is within the error bars of the current numbers in the paper, and above Prototypical networks trained and tested on 5-way (65.77 + / - 0.7)%. Further, this result was achieved quickly in response to this discussion without any tuning of the optimization hyper-parameters (learning rate etc.) to this new setting. Based on past experience with Versa, we are confident that this performance can be further improved.
> > > > >
> > > > > We will similarly investigate the performance in the 1-shot setting, and report final numbers for the next iteration of the paper. We would be happy to include the numbers achieved with both Versa training procedures above, as well as the numbers achieved by both training procedures achieved by Proto-nets in Table 3, along with a thorough discussion about the differences induced between the two procedures in the case of Versa and Prototypical networks.
> > > > >
> > > > > We do disagree with your statement that the two settings (5-way classification with 4 tasks per batch and 20-way classification) are the same. A 20-way classification task is not equivalent to four 5-way classification tasks, namely in the independence assumptions induced over the predictive distributions by the different normalizing constants. This is consistent with the fact that, in contradiction to what your claims would imply, Versa's performance on the meta-test set is not impacted by reducing the number of tasks per batch, while prototypical networks significantly benefits from higher-way training.
> > > > >
> > > > > Finally, several members of our team are away (or are about to depart) for the holidays. So, whilst we're very happy to discuss these matters further, we will only be able to do so after the Christmas break. We would be very happy for you to reach out to us directly at this time so that we can continue discussion in a more direct way with a quicker response time. Thank you again for your input, and happy holidays.

---

### Public Comment · (anonymous) · 2019-09-20
**Incorrect results for SNAIL?**

In your paper, you report 45.1% and 55.2% accuracies for SNAIL [1] for 1-shot and 5-shot, respectively, yet in their original paper they reported much higher results (55.71 ± 0.99% and 68.88 ± 0.92%). I could not find any explanations for the results you reported. Could you comment on it? Am I missing something? Thank you.

[UPDATE] I noticed the appendix in their paper where they report these results you reported for an architecture equivalent to the one used by other approaches. In any case, I think it's misleading to claim to be the state of the art on a dataset in this way. You could say at most that you are the state of the art on MiniImageNet with the Conv4 architecture.

[1] Nikhil Mishra, Mostafa Rohaninejad, Xi Chen, and Pieter Abbeel.  A simple neural attentive meta-learner. In International Conference on Learning Representations, 2018.
https://openreview.net/forum?id=B1DmUzWAW&noteId=B1DmUzWAW

---

### Meta-Review · Area_Chair1 · 2018-12-14

**Confidence:** 5
**Recommendation:** Accept (Poster)

**Metareview:**

The paper proposes a decision-theoretic framework for meta-learning. The ideas and analysis are interesting and well-motivated, and the experiments are thorough. The primary concerns of the reviewers have been addressed in new revisions of the paper. The reviewers all agree that the paper should be accepted. Hence, I recommend acceptance.